# Bioanalytical Assay Development and Validation for the Pharmacokinetic Study of GMC1, a Novel FKBP52 Co-chaperone Inhibitor for Castration Resistant Prostate Cancer

**DOI:** 10.3390/ph13110386

**Published:** 2020-11-13

**Authors:** Oscar Ekpenyong, Candace Cooper, Jing Ma, Naihsuan C. Guy, Ashley N. Payan, Fuqiang Ban, Artem Cherkasov, Marc B. Cox, Dong Liang, Huan Xie

**Affiliations:** 1Department of Pharmaceutical Sciences, College of Pharmacy and Health Sciences, Texas Southern University, Houston, TX 77004, USA; oscar.ekpenyong@yahoo.com (O.E.); candacejcooper1011@yahoo.com (C.C.); jing.ma@tsu.edu (J.M.); Dong.Liang@tsu.edu (D.L.); 2Department of Pharmacokinetics, Pharmacodynamics & Drug Metabolism, Merck & Co., Inc., South San Francisco, CA 94080, USA; 3Technical Services, Pfizer & Co., Inc., McPherson, KS 67460, USA; 4Department of Biological Sciences, University of Texas at El Paso, El Paso, TX 79968, USA; nguy08@gmail.com (N.C.G.); anpayan@miners.utep.edu (A.N.P.); mbcox@utep.edu (M.B.C.); 5Vancouver Prostate Centre, University of British Columbia, 2660 Oak Street, Vancouver, BC V6H 3Z6, Canada; fban@prostatecentre.com (F.B.); acherkasov@prostatecentre.com (A.C.)

**Keywords:** GMC1, bioanalytical method, LC-MS/MS, pharmacokinetics, prostate cancer, drug development

## Abstract

Background: GMC1 (2-(1H-benzimidazol-2-ylsulfanyl)-*N*-[(Z)-(4-methoxyphenyl) methylideneamino] acetamide) effectively inhibits androgen receptor function by binding directly to FKBP52. This is a novel mechanism for the treatment of castration resistant prostate cancer (CRPC). Methods: an LC-MS/MS method was developed and validated to quantify GMC1 in plasma and urine from pharmacokinetics studies in rats. An ultra-high-performance liquid chromatography (UHPLC) system equipped with a Waters XTerra MS C18 column was used for chromatographic separation by gradient elution with 0.1% (*v*/*v*) formic acid in water and methanol. A Sciex 4000 QTRAP^®^ mass spectrometer was used for analysis by multiple reaction monitoring (MRM) in positive mode; the specific ions [M+H]^+^
*m/z* 340.995 → *m/z* 191.000 and [M+H]^+^ m/z 266.013 → *m/z* 234.000 were monitored for GMC1 and internal standard (albendazole), respectively. Results: GMC1 and albendazole had retention times of 1.68 and 1.66 min, respectively. The calibration curves for the determination of GMC1 in rat plasma and urine were linear from 1–1000 ng/mL. The LC-MS/MS method was validated with intra- and inter-day accuracy and precision within the 15% acceptance limit. The extraction recovery values of GMC1 from rat plasma and urine were greater than 95.0 ± 2.1% and 97.6 ± 4.6%, respectively, with no significant interfering matrix effect. GMC1 is stable under expected sample handling, storage, preparation and LC-MS/MS analysis conditions. Conclusions: Pharmacokinetic evaluation of GMC1 revealed that the molecule has a biexponential disposition in rats, is distributed rapidly and extensively, has a long elimination half-life, and appears to be eliminated primarily by first order kinetics.

## 1. Introduction

Androgen receptor (AR)-regulated genes contribute to the initiation and progression of prostate cancer (PCa). Signaling via the AR axis is considered to be important in facilitating prostate carcinogenesis, although the precise mechanisms driving PCa are still being elucidated [1]. Current PCa therapies antagonize androgen receptors (AR) by competing for its ligand binding site [1,2,3,4]. However, these treatment options become ineffective once androgen-dependence is lost and PCa eventually moves into a castration resistance state [5,6,7]. Thus, there is an urgent need for new anti-androgens with entirely novel mechanisms of action. The 52 kDa FK506 binding protein (FKBP52) is a known positive regulator of AR as well as glucocorticoid (GR) and progesterone receptors (PR) and β-catenin. As such, FKBP52 represents a potential therapeutic target in PCa [2,8,9,10,11,12,13]. A series of small molecules that selectively inhibit FKBP52, mediated by enhancement of AR signaling by targeting the AR binding function 3 surface (BF3), have been identified and characterized [2,9,14,15]. More recently, GMC1, a new chemical entity, was also identified through structure-based in silico screening as a potential ligand for the FKBP52 peptidyl-prolyl-cis-trans-isomerase (PPIase) pocket [11,13,16]. It has been further demonstrated that GMC1 effectively inhibits AR and GR activity in a variety of cell lines [16]. The IC_50_ of GMC1 in MDA-kb2 cell AR reporter assays is 8.83 ± 3.55 µM, and in 52KO MEFs AR reporter assays is 0.66 ± 0.48 µM. Compared to previously reported FKBP52 inhibitors, GMC1 is predicted to simultaneously attenuate a variety of targets implicated in prostate cancer. This lead molecule effectively inhibits receptor activity in the low to mid micro-molar concentrations, AR-dependent prostate-specific antigen (PSA) secretion from a variety of prostate cancer cells, and 22Rv1 prostate cancer cell proliferation [2,16].

GMC1 is known by the IUPAC names 2-(1H-benzimidazol-2-ylsulfanyl)-*N*-[(Z)-(4-methoxyphenyl) methylideneamino] acetamide or (Z)-2-((1H-benzo[d]imidazol-2-yl) thio)-*N*′-(4-methoxybenzylidene) acetohydrazide. Figure 1 shows its chemical structure. Its molecular formula is C_17_H_16_N_4_O_2_S, and it has an average molecular weight of 340.4 g/mol. GMC1 has an acidic pKa of 10.61 ± 0.10 and basic pKa of 4.30 ± 0.10 at 25 °C. It has a calculated log*P* of 3.68 ± 0.57. The development of GMC1 as a drug is very appealing because it is a first-in-class molecule that directly targets the AR-associated co-chaperone for the treatment of castration-resistant prostate cancer (CRPC) [13]. For the preclinical and clinical development of GMC1, a suitable bioanalytical method for its quantification in in vitro and biological systems is highly essential. The sensitivity and selectivity of the bioanalytical method, as well as the stability of the analytes under storage conditions must be established in accordance with regulatory standards.

Herein, we report the development and validation of a simple, sensitive, and reproducible liquid chromatography-tandem mass spectrometry (LC-MS/MS) method for quantification of GMC1 in rat plasma and urine. We established the specificity, sensitivity and linearity, reproducibility, recovery, and dilution integrity of the assay in accordance with the United States Food and Drug Administration (FDA) bioanalytical method validation guidance [17]. Additionally, we assessed the stability of the analyte in samples under anticipated storage and handling conditions, as well as sample processing and analytical (LC-MS/MS) procedures. Following an in vivo pharmacokinetic study of GMC1 in adult male Sprague-Dawley (SD) rats, the fully validated assay method was deployed to determine the concentration of GMC1 in plasma and urine samples.

## 2. Results and Discussion

Validated analytical methods for quantification of drugs, their metabolites, and biomarkers are critical for successful drug discovery undertakings [4]. Herein, we report the development and full validation of a rapid, sensitive, and reproducible LC-MS/MS method for quantification of GMC1 in plasma and urine samples. The pharmacokinetic disposition of GMC1 in rats was subsequently evaluated using this analytical method. Our analytical method will therefore be very useful for studies related to the preclinical and clinical development of GMC1.

### 2.1. Chromatography and MS Conditions

The mobile phase, chromatographic column, column temperature, and source parameters were optimized for the signal intensity and peak shape. Symmetrical peak shape and optimal peak intensity were obtained with a Waters^®^ Xterra MS C_18_ column (2.1 × 50 mm, 3.5 µm), and 0.1% (*v*/*v*) formic acid in water and 0.1% (*v*/*v*) formic acid in methanol as aqueous and organic phases, respectively. Methanol as the organic phase yielded a better response compared to acetonitrile. Electrospray ionization (ESI) in positive mode was better suited for this analyte compared to atmospheric chemical ionization (APCI) and negative mode. Ion source parameters were optimized until the best intensity for the determination of the analyte was attained. The intensity of the analyte in samples prepared in mobile or organic phase containing formic acid seemed to diminish over time compared to samples prepared without any modifiers or additives. Hence standard solutions and protein precipitation solvent were devoid of formic acid.

The assay run time was 4.5 min, and the mean retention times for GMC1 and albendazole (internal standard (IS)) were 1.68 and 1.66 min, respectively. Chromatograms obtained from a zero-calibrator plasma sample, blank rat plasma and urine samples spiked with the lower limit of quantitation (LLOQ) level (1 ng/mL), and rat plasma sample spiked with IS are depicted in Figure 2. Carryover effect/ needle contamination were prevented by rinsing the needle (in-between each sample injection) with a strong organic wash solvent containing 60% isopropyl alcohol, 20% acetonitrile, and 20% methanol. No detectable GMC1 peak was observed in blank samples injected immediately after the upper limit of quantification (ULOQ) standard.

### 2.2. Method Validation

#### 2.2.1. Specificity and Selectivity

The specificity and selectivity of the assay (assessed to preclude interference from components of biological matrices and cross-reacting molecules) was evaluated by analyzing replicates of blank samples from different sources. As shown in Figure 2, the blank and zero calibrator samples were devoid of interfering signal at the observed retention times for both the analyte and internal standard. The instrument response for IS in blank rat plasma and urine samples was 2.9% and 3.3% of the average IS response for the standards and quality controls (QCs). In addition to specificity and selectivity, the observed IS response also indicates that the internal standard is stable when spiked into the precipitant solvent.

#### 2.2.2. Sensitivity and Linearity

The linear range for calibration curves for the determination of GMC1 in rat plasma and urine was from 1–1000 ng/mL, respectively, and the co-efficient for linear regression was accepted if greater than 0.99. The equations for the calibration curves in plasma and urine, respectively, were y = 9.89e^−4^x + 1.01e^−4^ and y = 1.54e^−3^x + 7.99e^−4^. The limit of detection (LOD) was based on a signal-to-noise (S/N) ratio of 3:1, while the LLOQ was selected based on an S/N ratio of at least 5:1. The measured LLOQ (1 ng/mL) of GMC1 in rat plasma and urine gave an S/N ratio much greater than five.

#### 2.2.3. Precision, Accuracy and Dilution Integrity

The precision and accuracy were expressed as the coefficient of variation (% CV) and the percentage relative error (% RE), respectively. The intra-day and inter-day accuracy and precision for all four QC samples were within the acceptable range: 20% for LLOQ and 15% for low, medium and high QC (LQC, MQC and HQC). Summarized in Table 1, the data confirm that GMC1 can be accurately and precisely quantified in rat plasma and urine from 1–1000 ng/mL using our LC-MS/MS method.

The dilution integrity was assessed by determining the precision and accuracy of samples diluted several folds with blank plasma. The accuracy and precision of the QC samples were well within 15% of the nominal concentration and presented CV < 15% (Table 2). The dilution integrity data suggest that the concentration of GMC1 in samples of greater concentration than the highest calibrator for this method can be accurately and precisely quantified following dilution with blank plasma and urine at least up to 50 times.

#### 2.2.4. Extraction Recovery

The ratio of the instrument response (PAR) from a sample spiked with GMC1 before protein precipitation to the mean instrument response (PAR) from a sample spiked with the analyte after the protein precipitation procedure, was calculated as the absolute recovery of GMC1 from biological matrix. Percentage recoveries > 95.0 ± 2.1% and 97.6 ± 4.6% from rat plasma and urine, respectively, were observed. The data (Table 3) indicate that GMC1 can be efficiently extracted from biological matrices by simple protein precipitation with methanol.

#### 2.2.5. Matrix Effect

The effect of biological matrix, expressed as matrix factor, indicates the propensity for analyte signal suppression or enhancement by co-eluting components of biological matrices during sample analysis. While a positive matrix factor signifies analyte signal enhancement, a negative matrix factor suggests analyte signal suppression [18]. The matrix effect is said to be significant if the calculated matrix factor is greater than ± 15%. The mean matrix factor, summarized in Table 3, suggests that there was no significant sample matrix effect interposing with the quantification of GMC1 in rat plasma and urine using our LC-MS/MS method [18].

Analyte signal suppression by co-eluting polyethylene glycol (PEG) 300 in the dosing solution for pharmacokinetic study was assessed by determining the accuracy and precision of QC samples containing 1% and 0.1% PEG 300. Accuracies of 107.7% and 108.4%, respectively, were observed for 0.1% PEG QC and 1% PEG 300 QC samples. Precisions for the PEG QC samples were 0.9% and 5.6%, respectively. The accuracy and precision were within 15% of the nominal concentration and presented CV < 15%, suggesting that analyte signal interference was not suppressed by PEG 300.

#### 2.2.6. Analyte Stability

The recovery of GMC1 from QC samples prepared and stored on the bench-top or at −80 °C, subjected to three freeze-and-thaw cycles, processed and placed on the LC system auto-sampler, were determined to assess the stability of the GMC1 under anticipated storage and handling conditions, as well as processing and sample analysis conditions. The results of the stability study (expressed as the average percent of initial GMC1 concentration remaining) are summarized in Table 4.

The bench-top stability data suggest that GMC1 is stable in rat plasma placed on the bench-top at room temperature for up to 4 h. The decreased percentage recovery at 6 h suggest that plasma samples should be immediately processed once thawed at room temperature. On the other hand, GMC1 is stable in rat urine at the same conditions for up to 6 h. The long-term storage stability was assessed in by analyzing QC samples stored at −80 °C for 14 days. Recoveries of 101.2 ± 0.8% and 97.0 ± 2.6%, respectively, in rat plasma and urine suggest that the analyte is stable when samples are frozen (at least for 14 days) before analysis. Mean recoveries from freeze-and-thaw cycled samples were greater than 94.1 ± 2.9% and 86.6 ± 7.5% for plasma and urine, respectively. This suggests that the analyte is stable in samples refrozen after being allowed to thaw at room temperature.

The effect of the IS on GMC1 recovery and stability was assessed by extracting the QC samples for an auto-sampler stability test with either pure methanol or methanol containing IS. The processed plasma and urine samples were stored on the LC system autosampler (temperature 10 °C) for up to 6 h before analysis. The average percentage recoveries, summarized in Table 4, suggest that GMC1 is stable in the extracted samples stored on the LC system for at least 6 h prior to analysis, and its stability is not affected by the IS.

### 2.3. Pharmacokinetic Study

The validated LC-MS/MS method was deployed to quantify GMC1 in rat plasma and urine samples from pharmacokinetic study of GMC1 in SD rats. Adult male SD rats were dosed a single 2 mg/kg i.v. bolus of GMC1. Plasma and urine samples collected from the study were analyzed using this assay method for GMC1 concentrations.

The plasma concentration—time profile, shown in Figure 3, illustrates that GMC1 is rapidly distributed after administration. Elimination appears to be slow after the rapid distribution phase. Table 5 shows the mean pharmacokinetic parameters describing the disposition of GMC1 in rats. In summary, following the administration of GMC1, an average maximum plasma concentration (C_max_) of 2.5 ± 0.6 mg/L was reached, rapidly declining within the first two h, and steadily tailing off in the terminal elimination phase. GMC1 has an elimination half-life of 14.8 ± 4.2 h, mean residence time of 10.7 ± 4.4 h and total plasma clearance of 1.7 ± 0.2 L/h/kg. Its volume of distribution in rats is greater than rat total body water (V_d_: 36.9 ± 9.7 L/kg). GMC1 is eliminated by mechanisms displaying first-order kinetics in rats. It appears to be extensively metabolized or eliminated by the biliary route. Only 0.01% of the total dose administered was excreted unchanged in urine 24 h after administration. 

## 3. Materials and Methods

### 3.1. Materials

GMC1, of purity ≥ 98%, was custom synthesized by ChemBridge™ (San Diego, CA, USA). Albendazole, formic acid, LC-MS grade methanol and water were purchased from Sigma Aldrich (St. Louis, MO, USA). Heparin sodium (1000 units/mL) and pharmaceutical grade normal saline (0.9% *v*/*v*) were purchased from Hospira Inc. (Lake Forest, IL, USA). Fresh rat plasma and urine were obtained from male SD rats purchased from Envigo RMS Inc. (Indianapolis, IN, USA), and stored at −80 °C until use.

### 3.2. Instruments and Conditions

Chromatographic separation was achieved with a Waters XTerra^®^ MS C_18_ column (2.1 × 50 mm, 3.5 μm, 125 Å, Milford, MA, USA) on a Shimadzu Nexera X2 UHPLC System (Columbia, MD, USA). A binary solvent system was used as mobile phase; Solvent A was 0.1% (*v/v*) formic acid in LC-MS grade water and Solvent B was 0.1% (*v/v*) formic acid in LC-MS grade methanol. Sample analysis was performed using a two-phase gradient elution set to 10% B from initial to 0.40 min, 35% B at 0.45 min, 95% B from 2.5 to 3.5 min, and 10% B from 3.55 to 4.5 min at a flow rate of 0.5 mL/min (Table 6). A 10 μL of protein precipitated sample was injected in each run.

A 4000 QTRAP^®^ mass spectrometer (MS) from AB Sciex (Foster City, CA, USA) which is a hybrid triple quadrupole linear ion trap (LIT) MS equipped with a Turbo V™ ion source, was used for mass spectral analysis. A GENIUS ABN2ZA Tri Gas Generator from Peak Scientific (Inchinnan, Scotland, UK) was used to generate pure nitrogen used as curtain gas, source and exhaust gases. The transition ions from a precursor ion ([M+H]^+^) to selected product ion for GMC1 *m*/*z* 340.995 → *m*/*z* 191.000) and internal standard *m*/*z* 266.013 → *m*/*z* 234.000) were detected in positive ionization mode by multiple reaction monitoring (MRM). Albendazole, a benzimidazole like GMC1, was selected as an internal standard (IS). The IonSpray voltage was set to 1100 V; the curtain gas was set to 25 psi and the collision CAD gas was set to high. The source heater was maintained at 700 °C with both the nebulizer gas and heater gas set to 50 and 30 psi, respectively. The collision energy was set at 27 eV for GMC1 and IS, respectively. Compound dependent parameters such as entrance potential (EP) and dwell time were optimized: EP of 10.00 V and dwell time of 50 ms for GMC1 and IS was employed. System control, acquisition data collection and processing were performed on the Analyst^®^ software v1.6.2 (Sciex, Foster City, CA, USA). The compound specific parameters are summarized in Table 7. The chemical structures of GMC1 and IS, and their MS fragmentation patterns are shown in Figure 1 and Figure 4.

### 3.3. Preparation of Standard and Quality Control Samples

Working stock solutions of GMC1 and IS, prepared in LC-MS grade methanol at 1 mg/mL and 0.5 mg/mL, respectively, were stored at −20 °C until use. Working standard solutions containing ten times the final desired concentrations of calibrators and quality control (QC) samples were prepared in 50% methanol in water. Blank, zero and non-zero calibrators were prepared in blank rat plasma and urine at GMC1 concentrations ranging from 1–1000 ng/mL. Quality control (QC) samples were also prepared in blank rat plasma and urine at LLOQ (1 ng/mL), low (LQC: 2.5 ng/mL), medium (MQC: 400 ng/mL) and high (HQC: 800 ng/mL) concentrations of GMC1. All plasma and urine samples were processed for LC-MS/MS analysis by simple protein precipitation using methanol. Briefly, 5 µL of blank or standard working solutions, respectively, were spiked onto 45 µL of rat plasma or urine. Then 200 µL of precipitant (methanol containing 25 ng/mL of IS) was added and vortexed for 30 s. The mixture was centrifuged for 10 min at 14,000 rpm and 4 °C. The recovered supernatant was then transferred to an auto-sampler vial for injection on the LC-MS/MS system.

### 3.4. Sample Preparation

Plasma and urine samples obtained from pharmacokinetic study subjects were stored at –80 °C until analysis. The plasma samples were prepared for LC-MS/MS analysis by a simple protein precipitation method, as described in the previous section. Briefly, 50 μL of plasma sample was extracted with 200 μL of precipitant (methanol containing 25 ng/mL of IS) followed by vortex mixing for 30 s. The mixture was then centrifuged for 10 min at 14,000 rpm and 4 °C. The recovered supernatant was then transferred to an auto-sampler vial for injection on the LC-MS/MS system. The rat urine samples were prepared in a similar manner.

### 3.5. Method Validation

A full validation of the LC-MS/MS method was performed according to the “US FDA, Center for Drug Evaluation and Research: Guidance for Industry—Bioanalytical Method Validation” [17]. The validated parameters were sensitivity and linearity, specificity and selectivity, accuracy and precision, dilution integrity, matrix effect, extraction recovery and analyte stability.

#### 3.5.1. Specificity and Selectivity

Blank rat plasma and urine samples from six different sources were analyzed to exclude any endogenous co-eluting interference or peaks close to the expected retention times of analytes and IS. The assay selectivity and specificity were also assessed by comparing the instrument response for the IS in blank samples to the average instrument response for IS in standards and QCs.

#### 3.5.2. Sensitivity and Linearity

The sensitivity of the assay was determined by analyzing six replicates of lower limit of quantification (LLOQ) QC in plasma and urine. Precision and accuracy ≥ 20% were considered acceptable to establish sensitivity. The LLOQ was based on a signal-to-noise (S/N) ratio greater than 5:1 while the limit of detection (LOD) was defined based on an S/N of 3:1.

Linearity was assessed by plotting calibration curves of GMC1 in rat plasma and urine, respectively. The calibration curves were constructed by plotting the peak area ratio (PAR) of GMC1 to IS against nominal concentrations of GMC1. Parameters such as slope, intercept and correlation coefficient of linear regression equation were estimated using least square regression analysis using a $1/x^{2}$ weighting.

#### 3.5.3. Precision, Accuracy and Dilution Integrity

Six replicates of LLOQ, LQC, MQC and HQC samples were analyzed using a calibration curve constructed on the same day to determine the intra-day accuracy and precision. The inter-day accuracy and precision was determined by analyzing six replicates of LLOQ QC, LQC, MQC and HQC samples using 3 different calibration curves constructed on three different days. The assay accuracy was established by the percentage relative error (% RE) from the nominal GMC1 concentrations, while the percentage coefficient of variation (% CV) was considered as the assay precision. The accuracy and precision in rat plasma and urine were determined in a similar manner.

The dilution integrity was established by determining the accuracy and precision of the measurement of rat plasma and urine samples with a GMC1 concentration 2.5X of the highest standard and diluted with up to 50 times to fall within the linear range of the assay. Rat plasma and urine samples containing 2500 ng/mL of GMC1 were prepared and diluted 5, 10, 20, and 50 times with blank rat plasma or 5, 12.5, 25 and 50 times with blank rat urine, respectively. This experiment was conducted in sextuplets. Following LC-MS/MS analysis, the concentrations of six replicates were corrected with the dilution factor and the percent accuracy and precision were then determined.

#### 3.5.4. Extraction Recovery

The extraction recovery was determined by analyzing triplicates of QC samples, rat plasma and urine spiked with the respective concentrations of GMC1, before or after protein precipitation. The percentage recovery of GMC1 was calculated using Equation (1):(1)$$\%\;\mathrm{Recovery}\;=\;\frac{{\mathrm{Response}}_{\mathrm{pre}-\mathrm{extraction}\;\mathrm{spike}\;\mathrm{sample}}}{{\mathrm{Response}}_{\mathrm{post}-\mathrm{extraction}\;\mathrm{spike}\;\mathrm{sample}}}\;\times \;100\%$$
where Response_pre-extraction spike sample_ is the average peak area count for GMC1 in plasma or urine sample spike with the analyte pre-protein precipitation, and Response_post-extraction spike sample_ is the average peak area count for samples spiked with GMC1 after protein extraction.

#### 3.5.5. Matrix Effect

The matrix effect was determined to assess the effect of components of biological matrices on the quantification of GMC1 in such biological matrices. A set of rat plasma and urine QC samples and QC samples in a neat solution (35% methanol) were analyzed to determine the matrix effect. Equation (2) was used to calculate the matrix factor:(2)$$\mathrm{Matrix}\;\mathrm{factor}\;(\%)\;=\left(\frac{{\mathrm{Response}}_{\mathrm{post}-\mathrm{extraction}\;\mathrm{spike}\;\mathrm{sample}}-{\mathrm{Response}}_{\mathrm{neat}\;\mathrm{sample}}}{{\mathrm{Response}}_{\;\mathrm{neat}\;\mathrm{sample}}}\;\right)\;\times \;100$$
where Response_post-extraction spike sample_ is the mean peak area for a sample spiked with GMC1 post protein precipitation, and Response_neat sample_ is the mean peak area for QC samples of same GMC1 concentration in matrix-free solution (35% methanol). 

Interference of analyte signal by co-eluting PEG, used as an excipient in dosing solutions for pharmacokinetic studies, is well documented [19,20]. Signal interference by PEG 300 used in this study was evaluated by determining the precision and accuracy of six replicates of QC samples prepared in rat plasma spiked with 0.1% PEG 300 and 1% PEG 300, respectively. The PEG QC samples were compared to plasma QC samples devoid of PEG 300.

#### 3.5.6. Analyte Stability

The stability of GMC1 in samples under anticipated storage and handling conditions, as well as sample preparation and LC-MS/MS analysis procedures was assessed by determining the percentage recovery of the analyte from samples subjected to short-term storage on the bench-top, freeze-and-thaw cycles, and processed and stored on the LC system auto-sampler prior to analysis. All the stability test experiments were conducted in triplicates.

Three sets of plasma and urine QC samples were left on the bench-top for 2, 4, and 6 h, respectively; each set was analyzed and compared to fresh samples of the same analyte concentration to establish the bench-top storage (short-term) stability of GMC1 in rat plasma and urine. The long-term stability of samples stored in frozen condition (−80 °C) before analysis was assessed by measuring the mean percentage recovery of GMC1 from rat plasma and urine samples stored for 14 days. The instrument response from the frozen samples was compared to the mean response from freshly prepared samples.

The stability of GMC1 in rat plasma and urine samples after freeze–thaw cycle was evaluated by analyzing QC samples exposed to three cycles of freeze (at −80 °C) and thaw (room temperature). The mean percentage recoveries from the samples were measured against fresh QC samples of the same GMC1 concentration. Additionally, GMC1 stability in processed (protein precipitated) samples stored on the autosampler before LC-MS/MS analysis was assessed by analyzing plasma and urine QC samples extracted with methanol and placed in the auto-sampler for 2 to 6 h prior to analysis. A set of the samples was precipitated with methanol containing IS, while the other set with plain methanol (without IS). Both sample sets were compared to fresh samples of similar GMC1 concentration. The temperature of the auto-sampler was set to 10 °C.

### 3.6. Pharmacokinetic Study

Adult male SD rats (body weight 300 to 350 g) were purchased from Envigo RMS, Inc, (Indianapolis, IN, USA) and kept in an environmentally controlled room (fed ad libitum) for at least one week before experiments. The protocol for the animal experiment was reviewed and approved by the Institutional Animal Care and Use Committee at Texas Southern University, and all animal experiments were conducted according to the National Institute of Health “Guide for the Care and Use of Laboratory Animals, 8th Edition” [21].

Jugular veins of the rats (*n* = 4) were cannulated under anesthesia one day prior to the study. A co-solvent formulation containing 10 mg/mL of GMC1 in 50% *v*/*v* of Labrasol^®^ and PEG 300, respectively, was prepared and diluted 10X with normal saline prior to administration. Each rat was administered a single 2 mg/kg intravenous (i.v.) bolus dose of the diluted GMC1 solution. About 250 μL serial blood was obtained from each rat prior to dosing and at 5, 15 and 30 min, 1, 2, 4, 6, 8, 12, 24 and 48 h after administration into heparinized blood sample collecting tubes. The blood was centrifuged to obtain plasma which was frozen at −80 °C until LC-MS/MS analysis. Urine from each rat was also collected for up to 24 h after dosing and also stored at the same condition until analysis. GMC1 concentration in the plasma and urine samples was determined within 48 h using our LC-MS/MS assay method.

### 3.7. Pharmacokinetic Analysis

Pharmacokinetic parameters for GMC1 were estimated using Phoenix WinNonlin v7.0 (Certara USA Inc., Princeton, NJ, USA). Non-compartmental analysis was performed to estimate the terminal elimination half-life (t_1/2_), total clearance (CL), area under the plasma concentration time curve (AUC_0–∞_), plasma concentration at time zero (C_0_), apparent volume of distribution (VD), volume of distribution at steady state (Vss), and mean residence time (MRT). The AUC_0−∞_ was calculated by a linear log trapezoidal (linear up log down) method. Uniform weighting scheme as well as 1/Y, 1/Y^2^, and 1/(Yhat)^2^ were compared for the most suitable weighting for the non-compartmental analysis. The optimal weighting was selected based on the correlation coefficient and observed vs. predicted fit of the plasma concentration vs. time plot. Based on these criteria, the uniform weighting scheme was most suited for the pharmacokinetic analysis on the individual profiles.

## 4. Conclusions

A simple, sensitive, and reproducible LC-MS/MS method was developed and fully validated for quantification of levels of the novel drug candidate, GMC1, in plasma and urine samples. This method was confirmed to be accurate and precise for the quantification of GMC1 in samples within the linear range of assay, from 1–1000 ng/mL of GMC1. The analyte stability in biological samples was not impacted by anticipated sample storage, handling, preparation, and LC-MS/MS analysis conditions, or the presence or absence of an internal standard and was also easily extractable from the biological matrices by simple protein precipitation. The pharmacokinetic data from this study revealed that GMC1 has a bi-phasic disposition in adult male SD rats. It is distributed extensively and is eliminated primarily by first order kinetics; it has a long plasma elimination half-life. This LC-MS/MS assay method will be very essential in subsequent studies of GMC1.

## 5. Patents

Cox, M.B.; Xie, H.; Ekpenyong, O. “Intravenous Formulation and LC/MS/MS Analysis Method for GMC1” US10,010,534, 3 July 2018.

## Figures and Tables

**Figure 1 pharmaceuticals-13-00386-f001:**
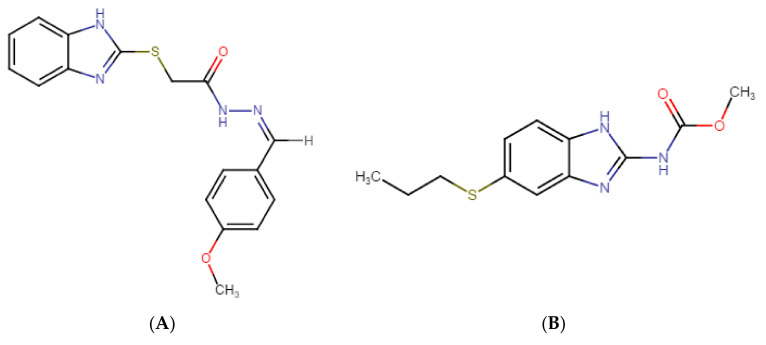
Chemical structures of (**A**) GMC1 (2-(1H-benzimidazol-2-ylsulfanyl)-*N*-[(Z)-(4-methoxyphenyl) methylideneamino] acetamide) (analyte) and (**B**) Albendazole (internal standard (IS)). Both compounds have a benzimidazole backbone. GMC1 has an acidic pKa of 10.61 ± 0.10 and a basic pKa of 4.30 ± 0.10 at 25 °C. It has a calculated log *p*-value of 3.68 ± 0.57 at room temperature.

**Figure 2 pharmaceuticals-13-00386-f002:**
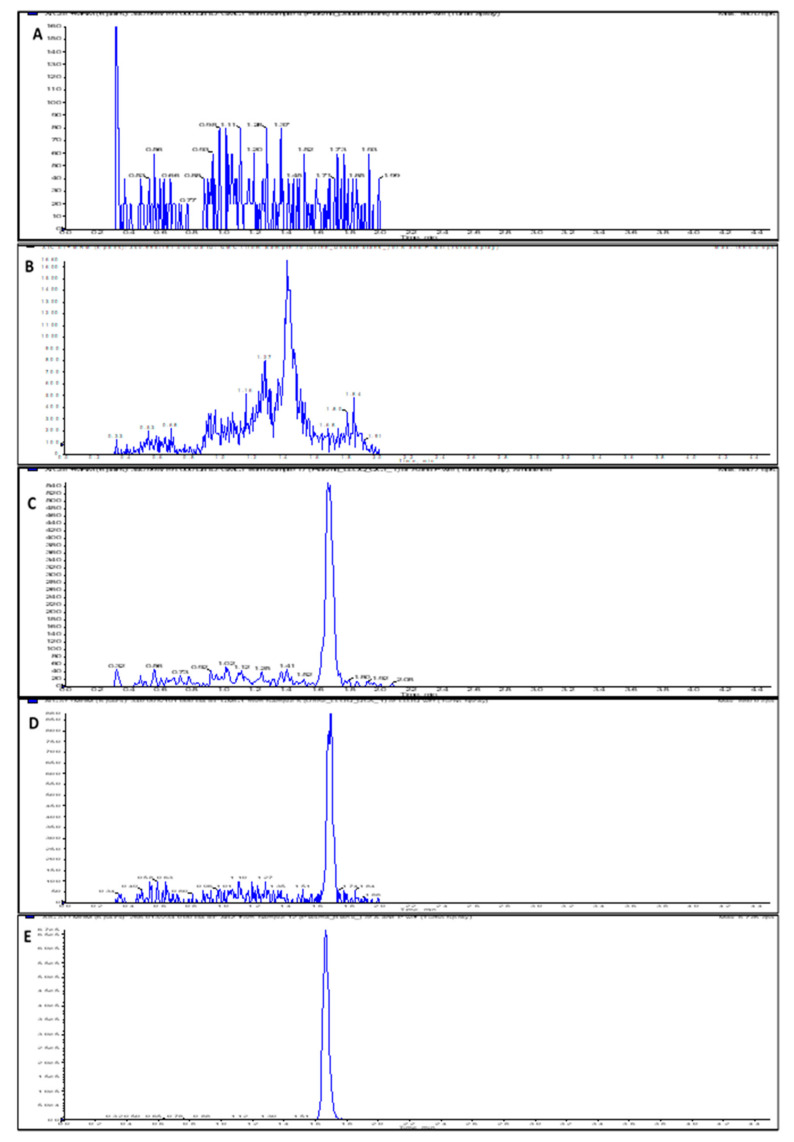
LC-MS/MS chromatograms (XIC of multiple reaction monitoring (MRM)) for (**A**) Double blank rat plasma sample; (**B**) Double blank rat urine sample; (**C**) Rat plasma sample spiked with LLOQ concentration (1 ng/mL) of GMC1; (**D**) Rat urine sample spiked with LLOQ concentration (1 ng/mL) of GMC1; (**E**) Blank rat plasma sample showing spiked IS.

**Figure 3 pharmaceuticals-13-00386-f003:**
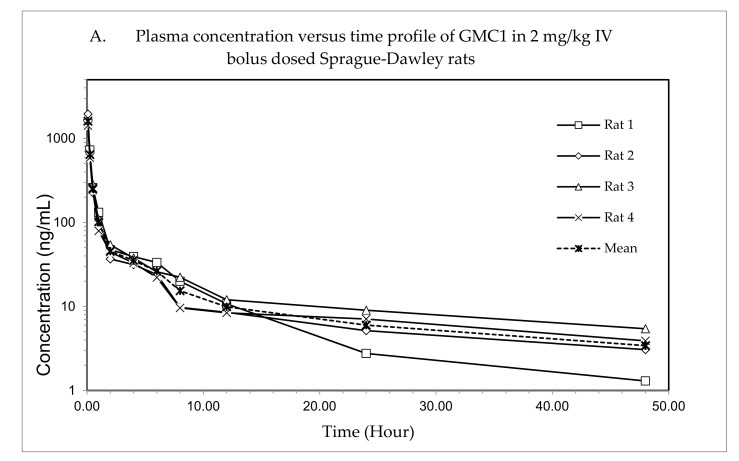
Pharmacokinetic study of GMC1 in adult male Sprague-Dawley (SD) rats (*n* = 4; average body weight = 0.33 ± 0.005 kg). (**A**) Plasma concentration—time profiles of GMC1 following a 2 mg/kg i.v. bolus dose, (**B**) Cumulative percentage of the administered dose of GMC1 excreted unchanged in urine.

**Figure 4 pharmaceuticals-13-00386-f004:**
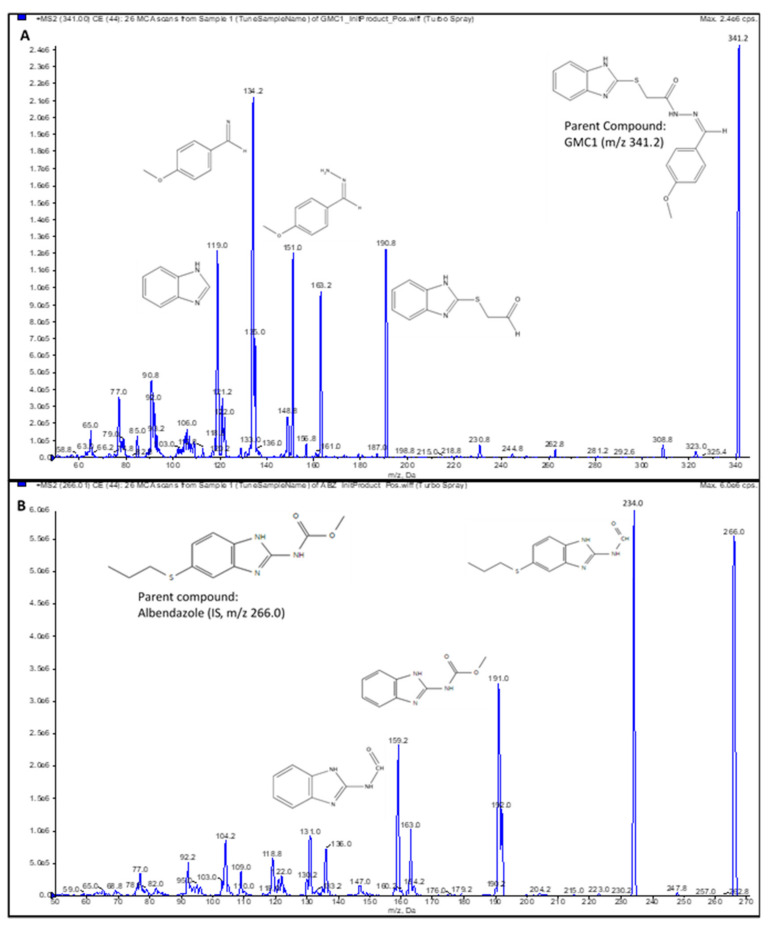
Product ion spectra. (**A**) GMC1 precursor ion (*m/z* 341.2) and product ions (*m/z* 190.8, 151.0 and 134.2); (**B**) Albendazole (IS) precursor ion (*m/z* 266.0) and product ions (*m/z* 234.0, 191.0 and 159.2). Their exact masses are 340.099 and 265.088, respectively. [M+H]^+^ ions were selected as precursor ions.

**Table 1 pharmaceuticals-13-00386-t001:** Accuracy and precision data of LC-MS/MS method for the quantification of GMC1 in various matrices. Percentage relative error (% RE) and coefficient of variation (CV) ≤ 20% for LLOQ and ≤ 15% for other QCs are considered acceptable.

Biological Matrix	QC	Nominal Concentration (ng/mL)	Intra-Day (*n* = 6)		Inter-Day (*n* = 6)	
			Accuracy (RE, %)	Precision (CV, %)	Accuracy (RE, %)	Precision (CV, %)
**Plasma**	**LLOQ**	1	96	9.35	93.7	8.4
	**LQC**	2.5	99.9	10.5	99.6	9.6
	**MQC**	400	90.2	7.1	92.8	2
	**HQC**	800	89	7.9	91.4	5.9
**Urine**	**LLOQ**	1	104.2	9.9	94	11.1
	**LQC**	2.5	101.1	7.6	93.2	8.4
	**MQC**	400	87.4	4.6	93.5	5.1
	**HQC**	800	88.6	7.5	92.2	5.5

**Table 2 pharmaceuticals-13-00386-t002:** Dilution integrity. Accuracy and precision demonstrate the effect of dilution on the quantification of GMC1 using this LC-MS/MS method (*n* = 6). RE and CV ≤ 15% are considered acceptable.

Biological Matrix	Dilution Factor	Accuracy (RE, %)	Precision (CV, %)
**Plasma**	5	104.9	6
	10	106.9	3.3
	20	89.9	5.6
	50	98.5	9.6
**Urine**	5	87.1	6.2
	12.5	97.3	5.6
	25	102	3.4
	50	88.4	2.6

**Table 3 pharmaceuticals-13-00386-t003:** Percentage extraction recovery and matrix factor for the quantification of GMC1 using this LC-MS/MS method (*n* = 3; (mean ± SD)). Matrix effect is considered significant if matrix factor is >±15%.

Biological Matrix	QC	Nominal Concentration (ng/mL)	Extraction Recovery	Matrix Factor
**Plasma**	LLOQ	1	98.8 ± 9.9	7.9 ± 3.3
	LQC	2.5	95.0 ± 2.1	7.2 ± 2.9
	MQC	400	98.3 ± 7.4	9.7 ± 2.9
	HQC	800	99.6 ± 4.8	6.7 ± 3.6
**Urine**	LLOQ	1	99.3 ± 4.4	5.1 ± 0.6
	LQC	2.5	97.6 ± 4.6	2.6 ± 1.1
	MQC	400	99.6 ± 6.8	6.5 ± 3.8
	HQC	800	98.2 ± 5.6	5.1 ± 2.9

**Table 4 pharmaceuticals-13-00386-t004:** Stability of GMC1 in samples for LC-MS/MS analysis (*n* = 3; mean (± SD)).

	Biological Matrix	Time	Mean Recovery ± SD (%)	
**Short-term/** **Long-term Stability**	Plasma	2 h	97.4 ± 8.9	
		4 h	88.9 ± 2.6	
		6 h	80.1 ± 5.7	
		14 d	101.2 ± 0.8	
	Urine	2 h	94.2 ± 4.7	
		4 h	93.9 ± 0.7	
		6 h	93.7 ± 1.2	
		14 d	97.0 ± 2.6	
**Processed sample** **Or Auto-sampler stability**		**Time(h)**	**Mean Recovery ± SD (%)**	
			**No IS**	**With IS**
	Plasma	2	94.7 ± 7.9	89.9 ± 4.7
		4	99.7 ± 4.6	97.5 ± 1.5
		6	98.4 ± 6.2	96.2 ± 5.6
	Urine	2	96.1 ± 2.4	96.4 ± 2.8
		4	89.7 ± 8.8	98.5 ± 1.1
		6	95.2 ± 2.1	97.1 ± 4.8
**Freeze-thaw Cycle** **Stability**		**Quality Control Sample**	**Mean Recovery ± SD (%)**	
	Plasma	LQC	106.0 ± 5.9	
		MQC	98.5 ± 3.7	
		HQC	94.1 ± 2.9	
	Urine	LQC	113.0 ± 8.9	
		MQC	101.0 ± 3.9	
		HQC	86.6 ± 7.5	

**Table 5 pharmaceuticals-13-00386-t005:** Pharmacokinetic parameters (mean ± SD) of GMC1 in SD rats following a single 2 mg/kg IV bolus dose of GMC1.

Pharmacokinetic Parameter (Unit)	Mean ± SD (*n* = 4)
C_0_ (mg/L)	2.5 ± 0.63
AUC_0__−∞_ (h*mg/L)	1.2 ± 0.12
T_1/2_ (h)	14.8 ± 4.2
VD (L/kg)	36.9 ± 9.7
Vss (L/kg)	18.3 ± 6.7
Cl (L/h/kg)	1.7 ± 0.2
MRT (h)	10.7 ± 4.4

C_0_ = concentration at time zero; AUC_0−∞_ = area under curve from time zero to infinity; T_1/2_ = elimination half-life; VD = apparent volume of distribution; Vss = volume of distribution at steady state; Cl = total clearance; MRT = mean residence time.

**Table 6 pharmaceuticals-13-00386-t006:** Gradient elution profile for the chromatographic separation of GMC1 and IS from matrices. Mobile Phase A—0.1% (*v*/*v*) formic acid in water and mobile phase B—0.1% (*v*/*v*) formic acid in methanol.

Time (min)	Flow Rate (mL/min)	Mobile Phase A (%)	Mobile Phase B (%)
**Initial**	0.5	90	10
**0.40**	0.5	90	10
**0.45**	0.5	65	35
**2.50**	0.5	5	95
**3.50**	0.5	5	95
**3.55**	0.5	90	10
**4.50**	0.5	90	10

**Table 7 pharmaceuticals-13-00386-t007:** Compound specific parameters for MS/MS acquisition of GMC1 and IS (albendazole).

Parent	Transition (*m/z*)	Dwell Time (msec)	DP (Volts)	EP (Volts)	CE (Volts)	CXP (Volts)
**GMC1**	340.995 → 191.000	50	81.0	10.0	27.0	10.0
**Albendazole (IS)**	266.013 → 234.000	50	86.0	10.0	27.0	14.0

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
