# Peer review of "Bioanalytical Assay Development and Validation for the Pharmacokinetic Study of GMC1, a Novel FKBP52 Co-chaperone Inhibitor for Castration Resistant Prostate Cancer"

_pharmaceuticals, 2020, doi:10.3390/ph13110386_

Round 1

Reviewer 1 Report

The manuscript entitled “Bioanalytical Assay Development and Validation for the Pharmacokinetic Study of GMC1, a Novel FKBP52 Co-chaperone Inhibitor for Castration Resistant Prostate Cancer” by Ekpenyong et al, aimed to design and validate a method for the determination and quantification of GMC1. This is a promising drug in the fight of resistant prostate cancer.

The experiments are well performed and the results interesting. The study revealed the biodistribution of GMC1 in a rat model.

Minor suggestions:

The authors did not explain the role of androgen receptors in the cancer initiation and progression. This aspect should be added in the introduction.

What is the effective concentration of drug?

In the abstract section the names of GMC1 and IS must be added.

Line 57. The chemical formula must have numbers in the form of subscripts.

The chemical structure of GMC1 must be added in the introduction.

Author Response

  1. The authors did not explain the role of androgen receptors in the cancer initiation and progression. This aspect should be added in the introduction.

RESPONSE: A short description has been added to the beginning of the Introduction, starting in Line 40.

  1. What is the effective concentration of drug?

RESPONSE: IC50 of GMC1 has been added to the Introduction, in Line 60-61.

  1. In the abstract section the names of GMC1 and IS must be added.

RESPONSE: Added the IUPAC name of GMC1 into abstract as suggested, in Line 18-19. IS (albendazole) was already mentioned in the original Abstract line 26.

  1. Line 57. The chemical formula must have numbers in the form of subscripts.

RESPONSE: Changed as suggested.

  1. The chemical structure of GMC1 must be added in the introduction.

RESPONSE: It is in Figure 1 and one sentence has been added to Line 68.

Reviewer 2 Report

This is a well written manuscript of definite scientific significance, and requires only a few alterations as itemized underneath.

line 19, change to "...This is a novel mechanism for the treatment..."

line 22, change to " ....by gradient elution with 0.1%..."

line 48, change to "....FKBP52 peptidyl-prolyl...."

line 56, change to ".... acetamide or (Z)....

line 97, change "rising" to "rinsing"

line 123, change "several folds" to "repeatedly"

line 132, change "urine respectively" to "urine, respectively, "

line 144, change "assess" to "assessed"

line 175, remove "the"

line 183, add "the" after "by"

line 255, change "benzimdazole" to "benzimidazole"

line 273, change "was" to "were"

line 309, change "day" to "days"

line 312, change "in the similar" to "in similar"

line 329, change "matrix" to "matrices"

line 349, change "in samples" to "of samples"

Author Response

  1. line 19, change to "...This is a novel mechanism for the treatment..."

RESPONSE:  Changed as suggested.

  1. line 22, change to " ....by gradient elution with 0.1%..."

RESPONSE: Changed as suggested.

  1. line 48, change to "....FKBP52 peptidyl-prolyl...."

RESPONSE: Changed as suggested.

  1. line 56, change to ".... acetamide or (Z)....

RESPONSE: Changed as suggested.

  1. line 97, change "rising" to "rinsing"

RESPONSE: Changed as suggested.

  1. line 123, change "several folds" to "repeatedly"

RESPONSE: We feel changing to “repeatedly” will change the meaning of the sentence, because we did dilute several folds to for this experiment. Therefore, we did not make this change.

  1. line 132, change "urine respectively" to "urine, respectively, "

RESPONSE: Changed as suggested.

  1. line 144, change "assess" to "assessed"

RESPONSE: Changed as suggested.

  1. line 175, remove "the"

RESPONSE: Changed as suggested.

  1. line 183, add "the" after "by"

RESPONSE: Changed as suggested.

  1. line 255, change "benzimdazole" to "benzimidazole"

RESPONSE: Changed as suggested.

  1. line 273, change "was" to "were"

RESPONSE: Changed as suggested.

  1. line 309, change "day" to "days"

RESPONSE: Changed as suggested.

  1. line 312, change "in the similar" to "in similar"

RESPONSE: Changed as suggested.

  1. line 329, change "matrix" to "matrices"

RESPONSE: Changed as suggested.

  1. line 349, change "in samples" to "of samples"

RESPONSE: Changed as suggested.

Reviewer 3 Report

The topic of the paper is interesting and is very valuable for future research of GMC1. The paper is written clearly and it is well organized.

There are minor issues that need to be addressed:

Line 177: it should be Table 7 instead of Table 6

Figure3: why do chromatograms show only analysis time until 2 min and not the rest up to 4.5 min?

line 251: it should be mass spectral analysis instead of chromatographic analysis because the section is about mass spectrometry and chromatography was discussed in the previous section

line 265: did the authors notice any issues in solubility of albendazole at such high concentrations?

Author Response

  1. Line 177: it should be Table 7 instead of Table 6

RESPONSE: Changed as suggested.

  1. Figure3: why do chromatograms show only analysis time until 2 min and not the rest up to 4.5 min?

RESPONSE: The chromatograms show the full 4.5 mins, not just 2 mins, the peaks eluted at 1.68 and 1.66 mins for analyte and IS respectively.

  1. line 251: it should be mass spectral analysis instead of chromatographic analysis because the section is about mass spectrometry and chromatography was discussed in the previous section。

RESPONSE: Changed as suggested.

  1. line 265: did the authors notice any issues in solubility of albendazole at such high concentrations?

RESPONSE: Albendazole was slightly soluble at 1 mg/mL in methanol, hence our reason for preparing the stock solution at 0.5 mg/mL. We did not notice any solubility issues at this concentration.